# FEDDISCRETE: A SECURE FEDERATED LEARNING ALGORITHM AGAINST WEIGHT POISONING

## ABSTRACT

Federated learning (FL) is a privacy-aware collaborative learning paradigm that allows multiple parties to jointly train a machine learning model without sharing their private data. However, recent studies have shown that FL is vulnerable to availability poisoning attacks, integrity backdoor attacks and inference attacks via weight poisoning and inference. In this paper, we propose a probabilistic discretization mechanism on the client side, which transforms the client's model weight into a vector that can only have two different values but still guarantees that the server obtains an unbiased estimation of the client's model weight. We theoretically analyze the utility, robustness, and convergence of our proposed discretization mechanism and empirically verify its superior robustness against various weight-based attacks under the cross-device FL setting.

## 1 INTRODUCTION

Federated learning (FL) is an emerging privacy-aware framework that trains a machine learning model across multiple parties without accessing their local private data (Konečnỳ et al., 2016; McMahan et al., 2017). In FL, each client first trains the local model using its private data and then sends the *model gradients* to an honest central server. The central server aggregates all local model gradients to form a global model, which is sent back to the clients for the next round of training. However, sharing model gradients might still leads to security concerns and privacy leakage.

Recent studies have shown that FL is vulnerable to various model weight poisoning attacks. The adversarial client(s) can stealthily manipulate the global model via modifying the local model updates to achieve attack goals like, preventing model convergence, implanting backdoors into the global model, and inferring the privacy information of clients' private data (Gu et al., 2017; Blanchard et al., 2017; Pyrgelis et al., 2017; Xie et al., 2019; Bagdasaryan et al., 2020; Wang et al., 2020; Tang et al., 2020). To address such an issue, recent works start to explore different defense techniques to robustify FL against model weight poisoning attacks (Blanchard et al., 2017; Bagdasaryan et al., 2020; Shen et al., 2016; Geyer et al., 2017; Fung et al., 2018). However, most existing works apply the defense or robust techniques at the server side, which changes the role of the central server from being *honest* to *trusted*. And prior approaches mostly only focus on one type of attacks.

**Our contributions.** In this paper, we propose the FEDDISCRETE, a flexible FL framework that can combine any popular FL algorithms, for example, FedAvg(McMahan et al., 2017) and FedProx(Li et al., 2020), with our probabilistic discretization mechanism. Theoretical analyses on the utility, robustness, and convergence are performed to show that FEDDISCRETE is inherently robust to availability poisoning attacks (Kurita et al., 2020), integrity backdoor attacks (Gu et al., 2017) and server inference attacks (Shokri et al., 2017). FEDDISCRETE is evaluated on four popular image classification datasets under both data are i.i.d and non-i.i.d settings. The numerical results indicate that FEDDISCRETE is robust to various weight-based attacks.

## 2 BACKGROUNDS AND RELATED WORKS

In this section, we provide necessary background information on concepts thorough the paper and formalize the problem to be solved.

**Federated learning (FL).** FL is a privacy-aware collaborative learning, where $N$ clients and one trusted server work together to learn a global model (McMahan et al., 2017). Depending on the application scenarios, the number of clients $n$ can range from as small as two to several hundreds in the *cross-silo* setting or can easily go beyond millions in the *cross-device* setting. A classical way of formulating FL into an optimization problem is (Wang et al., 2021a)

$$\min_{\boldsymbol{w} \in \mathbb{R}^d} F(\boldsymbol{w}) := \sum_{i=1}^{N} p_i F_i(\boldsymbol{w}), \text{where } F_i(\boldsymbol{w}) = \frac{1}{|D_i|} \sum_{\xi \in D_i} f_i(\boldsymbol{w}; \xi) \text{ and } \sum_{i=1}^{N} p_i = 1. \tag{1}$$

In Eq. 1, $\boldsymbol{w}$ is the global model weight; $F_i$ is the $i$-client local objective function; the local loss functions $f_i(\boldsymbol{w}; \xi)$ are often assumed to be the same across all clients; the local data $D_i$ can have different distribution. Following the seminal work of (McMahan et al., 2017), extensive researches have been conducted to address various challenges in FL. For instance, (Li et al., 2020; Reddi et al., 2020) aim to design efficient optimization algorithms; (Konečný et al., 2016; Luping et al., 2019) try to improve the communication efficiency; (Bagdasaryan et al., 2020; Xie et al., 2021) study the security issues on both attacks and defenses.

**Adversaries.** For any FL attack, the server is assumed to be honest, which faithfully follow the training protocols and any client cannot directly get the model weight of other clients. Under these assumptions, there are two adversaries: 1) the malicious clients can manipulate the weights for various attack purposes, such as interfering the global model training or implanting backdoors into the global model; 2) the curious server can explore the privacy information from local clients via inference attacks.

**Availability poisoning attacks.** The goal of availability poisoning attacks (APAs) is for malicious local client(s) to destroy or reduce the global model's utility (Shen et al., 2016; Sun et al., 2018). In APAs, the attacker can either control the global model's utility on target tasks or decrease the most of the model's utility as the optimal attack strategy. In practice, adding a large random noise can successfully worsen the global model's utility but can be easily discovered by anomaly detection techniques. In this case, any advanced attack needs to know and tries to bypass the defense techniques used in training. In this work, we assume the attacker has full knowledge of training process and all possible defense techniques.

**Integrity backdoor attacks.** The goals of integrity backdoor attack (IBAs) are two-fold: 1) the attacker aims to implant backdoors into the global model, through which the attacker can control the prediction results by injecting the trigger to any clean examples; 2) at the same time, the attacker wants the global model can still perform as good as the non-attacked model on all clean inputs (Bagdasaryan et al., 2020). Note that, IBAs are also conducted by the malicious clients in FL.

**Inference attacks.** Many studies show that the attacker could explore the private information from the model weights, such as membership and attribute information (Pyrgelis et al., 2017; Shokri et al., 2017; Ganju et al., 2018; Melis et al., 2019) or even recover the training samples used by clients Zhu et al. (2019). When the honest-but-curious server becomes malicious, it can explore privacy information of each client's local dataset through inference attacks (IAs) since the server has full access to clients' model weight.

Both APAs and IBAs can be achieved via weights poisoning that modifies the local model weights. For example, as the global model converges, the deviations of local models start to cancel out, i.e., $\sum_{i=1}^{n_t-1}(\boldsymbol{w}_i^t - \boldsymbol{w}^{t-1}) \approx 0$ given as in Eq. 2 of Bagdasaryan et al. (2020), where $t$ and $i$ the communication round and client index respectively, $\boldsymbol{w}^t$ and $\boldsymbol{w}_i^t$ are the global and local model respectively, and $n_t$ is the number of participating clients. If the adversary aims to replace global model $\boldsymbol{w}^t$ with a target model $X$, then it could propose to upload a local model weight $\boldsymbol{w}_A^t = n_t X - (n_t - 1)\boldsymbol{w}^{t-1} - \sum_{i=1}^{n_t-1}(\boldsymbol{w}_i^t - \boldsymbol{w}^{t-1}) \approx n_t(X - \boldsymbol{w}^{t-1}) + \boldsymbol{w}^{t-1}$ as given in Eq. 3 in Bagdasaryan et al. (2020). Prior works (Bagdasaryan et al., 2020; Wang et al., 2020; Xie et al., 2019) demonstrate the effectiveness of both APAs and IBAs when the training algorithm is FedAvg. IAs can be easily achieved by many existing advanced attack methods once they have the full access to the client model Shokri et al. (2017); Hu et al. (2021).

**Related works.** To address the various attacks, many prior defense works have been proposed and achieved good performance against the attacks. However, prior defense works only address one aspects of aforementioned three attacks. For *availability poisoning attacks*, Steinhardt et al. (2017) proposed a defending framework, which can be applied to defenders that remove outliers and then

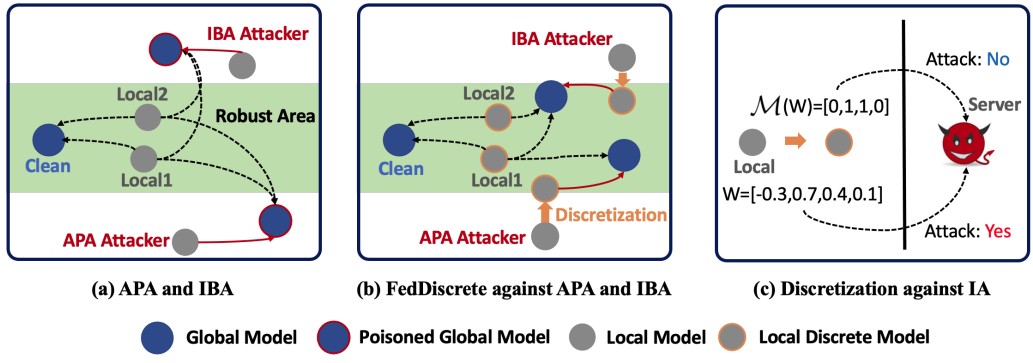

Figure 1: (a): An illustration on how adversarial clients can do APAs and IPAs via weight modification. If there is no adversaries, then FL training algorithm will give a clean model. When there are attackers, the clean model becomes poisoned. (b): An illustration on how discretization mechanism helps to defend the APAs and IPAs, which brings the poisoned model to the robust area. The robust area is a parameter space where the model would have the same performance as the clean model. (c) An illustration on how discretization mechanism helps to defend the IAs as the discrete model updates provide limited information compared to the continuous counterparts.

minimize a margin-based loss on the remaining data. Under assumptions, this framework provides the approximate upper bounds on the efficacy of any data poisoning attack. However, this framework does not fit in the FL setting, as it needs the access to all private data. Blanchard et al. (2017) studied the presences of Byzantine adversaries and proposed Krum aggregation rule to defend under the assumption that the participants' training data are i.i.d, which is not necessarily true in the FL setting. And indeed as argued in (Bagdasaryan et al., 2020), Krum can be used by the adversaries to make the attack more effective. FoolsGold (Fung et al., 2018) can mitigate sybils data poisoning attacks under the assumption that the honest clients can be separated from sybils by the diversity of their gradient updates. As discussed by the authors, FoolsGold is not successful at mitigating attacks initiated by a single adversary. For *integrity backdoor attacks*, Xie et al. (2021) provided a general framework that is certifiably robust to the backdoor attacks under the FL setting via the model weight smoothing. However, the convexity assumption is imposed on the loss function, which limits certified robustness to more challenging and widely used deep neural network. Ozdayi et al. (2021) proposed a defense approach based on adjusting server's learning rate with the guidance of sign information of agents' updates. For *inference attacks*, differential privacy is widely adopted approach to defend, where are judiciously random noise added on either the clients' model update or the global model, e.g., Geyer et al. (2017), but it suffers from the trade-off between privacy and accuracy. Compared to this work, neither of these works can address three types of attacks simultaneously.

## 3 METHOD

We introduce FEDDISCRETE, a federated learning framework with a simply yet effective discrete mechanism and the flexibility to accommodate various FL optimization algorithms. A complete description is given in Algorithm 1. FEDDISCRETEdifferences FEDDISCRETE consists of three stages, namely, local training, discretization and aggregation, which are discussed in details.

**Local Training.** Since we mainly focus on the *cross-device* setting, in each communication round $t$, the server first selects a subset of $|S_t| = K$ clients to participate the current round of training and broadcasts the global model $\boldsymbol{w}^t$ to the selected clients. The clients who receives the global model then performs local training using any appropriate algorithm to generate a new local model, i.e. $\boldsymbol{w}_i^{t+1}$ for all $i \in S_t$. The $i$th client can 1) use the local stochastic gradient descent(SGD) method as is considered in FedAVG to perform a fixed number of SGD steps to improve communication efficiency; 2) use adaptive method (Wang et al., 2021b) to improve the convergence; 3) approximately minimize $F_i(\boldsymbol{w}) + \frac{\mu}{2}\|\boldsymbol{w} - \boldsymbol{w}^t\|^2$ for some $\mu > 0$ as proposed in FedProx to accommodate the system and data heterogeneity (Li et al., 2020).

**Discretization.** For all $i \in S_t$, the $i$th local client computes the maximum and minimum values of $\boldsymbol{w}_i^t$ and sends noise-perturbed maximum value $u_i^t$ and minimal value $l_i^t$ back to the server, where the

noise is added to protect the privacy of the local training data and the noise is sampled and added in a way that to guarantee $u_i^t$ and $l_i^t$ are the valid upper bound and lower bound of $\boldsymbol{w}_i^{t+1}$ element-wise. Once the server received the client-wise upper-bounds and lower-bounds, it computes the minimum of the lower bounds $l_{\min}^t$ and the maximum of the upper bounds $u_{\max}^t$ to prepare the inputs for the discretization mechanism (see Definition 3.1), which outputs an unbiased estimator of the input (see Lemma 4.1). For all $i \in S_t$, the $i$th client discretizes its continuous model $\boldsymbol{w}_i^t$ into $\mathcal{M}(\boldsymbol{w}_i^t; l_{\min}^t, u_{\max}^t)$ and uploads $\mathcal{M}(\boldsymbol{w}_i^t; l_{\min}^t, u_{\max}^t)$ to the server for aggregation.

**Aggregation.** Rather than simply aggregating $\{\mathcal{M}(\boldsymbol{w}_i^t; l_{\min}^t, u_{\max}^t)\}_{i \in S_t}$, the sever first inspects whether each coordinate of $\boldsymbol{w}_i^t$ is either $l_{\min}^t$ or $u_{\max}^t$ to exclude the any potential malicious adversaries that attempt to bypass the discretization process. And the server only aggregates the local model weights that pass the sanity check. Also when the server computes $l_{\min}^t$ and $u_{\max}^t$, it can be more conservative by discarding the extreme values in $\{l_i^t\}_{i \in S_t}$ and $\{u_i^t\}_{i \in S_t}$ by setting thresholds on the lower quantile of $\{l_i^t\}_{i \in S_t}$ and upper quantile of $\{u_i^t\}_{i \in S_t}$ to prevent adversaries proposing extremely large upper bounds and/or small lower bounds. For the ease of presentation, we do not include this choice in the description of Algorithm 1.

**Definition 3.1** (Discretization Mechanism). For any $(w, l, u) \in \mathbb{R} \times \mathbb{R} \times \mathbb{R}$ with $l \le w \le u$, define the discretization mechanism as

$$\mathcal{M}(w; l, u) = \begin{cases} u, & \text{w.p. } \frac{w-l}{u-l} \\ l, & \text{w.p. } \frac{u-w}{u-l}, \end{cases} \tag{2}$$

where w.p. is the shorthand notation for "with probability".

---

**Algorithm 1** FEDDISCRETE

---

1: **Input:** Initial model weight $\boldsymbol{w}^0 \in \mathbb{R}^d$, total training rounds $T$, the participants size $K$, and the learning rate $\{\eta^t\}_{t=0}^{T-1}$ and a positive sequence $\{\sigma_i\}_{i=1}^N$.
2: **for** $t = 0, 1, 2, \ldots, T-1$ **do**
3:     The sever randomly selects an index set $S_t$ with $|S_t| = K$ and broadcasts $\boldsymbol{w}^t$ to the client $i$ if $i \in S_t$.

4:     For the $i$th client, where $i \in S_t$, it performs the local training to obtain $\boldsymbol{w}_i^{t+1}$ and samples two random variables $\xi_i^t$ and $\zeta_i^t$ from the truncated Gaussian $\text{TN}(0, \sigma_i; 0, 1)$. Then it computes and uploads $l_i^t = \min_{j \in [d]}\{[\boldsymbol{w}_i^t]_j - \boldsymbol{w}^t\} - \xi_i^t$ and $u_i^t = \max_{j \in [d]}\{[\boldsymbol{w}_i^t]_j - \boldsymbol{w}^t\} + \zeta_i^t$ to the server.

5:     The sever computes $l^t = \min_{i \in S_t}\{l_i^t\}$ and $u^t = \max_{i \in S_t}\{u_i^t\}$ then broadcasts $l^t, u^t$ to clients in $S_t$.

6:     For the $i$th client, where $i \in S_t$, it applies the discretization mechanism $\mathcal{M}$ is an element-wise fashion and uploads the discrete model weight $\mathcal{M}(\boldsymbol{w}_i^{t+1} - \boldsymbol{w}^t; l^t, u^t)$ to the server.

7:     The server conducts the sanity check for $\{\mathcal{M}(\boldsymbol{w}_i^{t+1} - \boldsymbol{w}^t; l^t, u^t)\}_{i \in S_t}$, i.e., validates whether all elements of $\boldsymbol{w}_i^{t+1}$ are either $l^t$ or $u^t$. Form the set $S_t' \subseteq S_t$ to collect all the clients' index passing the sanity check and compute $\boldsymbol{w}^{t+1} = \boldsymbol{w}^t + \eta^t \frac{1}{|S_t'|} \sum_{i \in S_t'} \mathcal{M}(\boldsymbol{w}_i^{t+1} - \boldsymbol{w}^t; l^t, u^t)$.

8: **end for**

---

We close this section by making following comments.

- Discretization is a widely applied technique, for example, in influence maximization Kempe et al. (2003), the algorithm needs to simulate the propagation from a probabilistic graph that is consists of a set of edges with activation probabilities in $[0, 1]$. It is also can be regarded as a special form of quantization that is widely used in the distributed optimization (Alistarh et al., 2017; Reisizadeh et al., 2020) to improve communication efficiency. Compared with prior discretization works, this is the first work to adopt discretization to protect FL from weight poisoning attack.

- Compared with the FedAvg, although an additional round of communication is required in FEDDISCRETE to perform the discretization mechanism. We argue that FEDDISCRETE is more communication-efficient than FedAvg. Assume each scalar takes 64bits, then for the

$t$-th round, the total bits communicated are $128dK$ for FedAvg, while for FEDDISCRETE are $64dK + 128K + dK$. For large $d$, FEDDISCRETE communicates less bits. [1]

- Although the discretization mechanism outputs an unbiased estimation of $\boldsymbol{w}_i^{t+1}$, the variance of the estimator $\mathcal{M}(\boldsymbol{w}_i^{t+1})$ can be large. To mitigate this issue, one can compute more finely-grained upper bounds and lower bounds. For example, assume each client wants to train a neural network. Then, instead of computing the maximum and minimum values over the entire $\boldsymbol{w}$, the client could compute layer-wise maximum and minimum values. And the discretization mechanism could be applied layer-wise with tighter upper bounds and lower bounds. This leads to smaller variance in the estimator at the cost of increasing the communicated bits.

- Intuitively, the discretization mechanism is effective in defending weight poisoning attacks as the judiciously crafted adversary model is discretized. For example, consider the model replacement attack in Eq. 3 of Bagdasaryan et al. (2020), once the malicious model $\boldsymbol{w}_A^t$ is discretized into $\mathcal{M}(\boldsymbol{w}_A^t)$, the model replacement becomes ineffective.

## 4 THEORETICAL ANALYSIS

In this section, we analyze the utility of the discretization mechanism and the convergence property of the FEDDISCRETE. Due to limited space, complete proofs are deferred to the appendix.

We first introduce the notations are used throughout the paper. $[\boldsymbol{w}]_j$ denotes the $j$th coordinate of the vector $\boldsymbol{w}$. Unless specified otherwise, $\|\cdot\|$ and $\|\cdot\|_\infty$ are the Euclidean norm and infinity norm respectively. To avoid the cluttered notation, we use $\mathcal{M}(\cdot)$ instead of $\mathcal{M}(\cdot;\cdot,\cdot)$ when there is no confusion. And if $\mathcal{M}$ is applied over a vector, we mean to apply it in a element-wise fashion. Here, $[d]$ represents $\{1, \cdots, d\}$.

### 4.1 UTILITY ANALYSIS

The first lemma shows that the discretization mechanism produces an unbiased estimator with bounded variance.

**Lemma 4.1.** *For any $\boldsymbol{w} \in \mathbb{R}^d$ and $(l, u) \in \mathbb{R} \times \mathbb{R}$ such that $l \leq \min_{j \in [d]} [\boldsymbol{w}]_j < \max_{j \in [d]} [\boldsymbol{w}]_j \leq u$, then*

- $\mathbb{E}[\mathcal{M}(\boldsymbol{w}; l, u)] = \boldsymbol{w}$;

- $\mathbb{E}[\|\mathcal{M}(\boldsymbol{w}; l, u) - \boldsymbol{w}\|^2] \leq \frac{d(u-l)^2}{4}$ *and* $\mathbb{E}[\|\mathcal{M}(\boldsymbol{w}; l, u) - \boldsymbol{w}\|^2] \leq \frac{\|\boldsymbol{w}\|^2}{4}$.

A natural question to ask is that how far is the averaged local model weights $\bar{\boldsymbol{w}}^t := \sum_{i \in S_{t-1}} \boldsymbol{w}_i^{t-1}$ from its discretized counterpart $\bar{\boldsymbol{w}}_{\mathcal{M}}^t := \sum_{i \in S_{t-1}} \mathcal{M}(\boldsymbol{w}_i^{t-1})$. The next theorem characterizes the distance between $\bar{\boldsymbol{w}}^t$ and $\bar{\boldsymbol{w}}_{\mathcal{M}}^t$.

**Theorem 4.2.** *For any communication round $t \leq T$ and any $\beta \in (0, 1)$, there exists $\epsilon^t = \mathcal{O}\left(\frac{(u^t - l^t)\sqrt{\log \frac{2d}{\beta}}}{\sqrt{K}}\right)$ such that*

$$\mathbb{P}\left(\left\|\bar{\boldsymbol{w}}^t - \bar{\boldsymbol{w}}_{\mathcal{M}}^t\right\|_\infty \leq \epsilon^t\right) \geq 1 - \beta.$$

### 4.2 ROBUSTNESS ANALYSIS

Next, we show the discrete mechanism is robust to random weight poisoning and can recover the original weight from local models against a limited number of independent targeted weight poisonings. To maximize the attack impact, the attacker choose to inverse the discrete local updates to

---

[1]Each model weight takes $64d$ bits. For FedAvg, server broadcasts the global model weight and clients upload the local model weights, so the total bits are $128dK$. For FEDDISCRETE, $64dK$ is the cost for the server broadcasting the global model to $K$ server. $128K$ is the cost for uploading and broadcasting the upper bounds and lower bounds. Since the discretized model weight $\mathcal{M}(\boldsymbol{w}_i^{t+1})$ can only taking two values, the clients can indeed just upload $d$ bits string and the server can recover the $\mathcal{M}(\boldsymbol{w}_i^{t+1})$.

attack the global model. If the intend update of the $j$th weight $[\boldsymbol{w}]_j$ is $u$, the attacker uploads $l$ to the server, and vice versa.

**Theorem 4.3.** *Suppose there are $F$ attackers out of total $N$ clients and denote $\bar{\boldsymbol{w}}_{\mathcal{M}} = \frac{1}{N}\sum_{i=1}^{N}\boldsymbol{w}_i$, then for any $j \in [d]$,*

$$[\mathbb{E}[\bar{\boldsymbol{w}}_{\mathcal{M}}]]_j = \frac{1}{N}\left(\sum_{i=1}^{N-F}[\boldsymbol{w}_i]_j - \sum_{i=N-F+1}^{N}[\boldsymbol{w}_i]_j\right) + (h+l)\frac{F}{N}. \tag{3}$$

More importantly, FEDDISCRETE has an asymptotically non-bias estimation when the number of clients approaches to $+\infty$ with a fixed number of attackers.

**Corollary 4.3.1.** *Let $N$ be $+\infty$, for some fixed $F$ attackers, the expectation of the averaged discrete weights is $\mathbb{E}[\bar{\boldsymbol{w}}_{\mathcal{M}}] = \boldsymbol{w}$.*

**Discussion.** The above theorem and corollary have shown the robustness against the weights poisoning attacks, such as IBAs and APAs. Besides, another important reason is the discretization mechanism limits the modification ability of the adversaries as shown in Figure 1. For example, if the clean aggregated model's weight is $[0.1, -2.3, 3.8]$, the IBA or APA could work while modifying the weight from $[0.1, -2.3, 3.8]$ to $[1, 0.2, 0.4]$, where the weight difference between clean and poisoned model is $[0.9, -2.5, -3.4]$. However, due to the discretization mechanism, while we only have 1 attacker among 100 clients, its modification could only be 0 or $\pm 0.01$ with the upper bound $u = 1$ and the lower bound $l = 0$. In this case, the adversary can not achieve its goal in any manner.

## 4.3 CONVERGENCE ANALYSIS

In this section, we prove the convergence of Algorithm 1 by instantiating the optimization algorithm for generating local model weights as the FedProx(Li et al., 2020). Specifically, at the $t$th round, in line 4 of Algorithm 1, the $i$th selected client produce the local model weight $\boldsymbol{w}_i^{t+1} \approx F_i(\boldsymbol{w}) + \frac{\mu}{2}\|\boldsymbol{w} - \boldsymbol{w}^t\|^2$ in the sense that $\left\|\nabla F_i(\boldsymbol{w}_i^{t+1}) + \mu(\boldsymbol{w}_i^{t+1} - \boldsymbol{w}^t)\right\| \leq \gamma \|\nabla F_i(\boldsymbol{w}^t)\|$ for some properly chosen $\mu > 0$ and $\gamma > 0$. The Assumption 4.1 is made throughout the whole section.

**Assumption 4.1.**

1. (Smoothness) For all $i \in [N]$, $F_i(\boldsymbol{w})$ is $L$-smooth, i.e., $\|\nabla F_i(\boldsymbol{w}) - \nabla F_i(\boldsymbol{w}')\| \leq L\|\boldsymbol{w} - \boldsymbol{w}'\|$ for some $L > 0$ and all $(\boldsymbol{w}, \boldsymbol{w}') \in \mathbb{R}^d \times \mathbb{R}^d$.

2. (Lower-bounded eigenvalue) The minimal eigenvalue of the Hessian of the client loss function $\nabla^2 F_i(w)$ is uniformly bounded below by a constant $\lambda_{\min} \in \mathbb{R}$.

3. (Bounded dissimilarity) For all $i \in [N]$ and any $\boldsymbol{w} \in \mathbb{R}^d$, $\mathbb{E}_i[\|\nabla F_i(\boldsymbol{w})\|^2] \leq B^2 \|\nabla f(\boldsymbol{w})\|^2$, where the expectation is taken with respect to the client index $i$.

4. (Algorithmic choices) In Algorithm 1, in line 3, the $i$th client is picked with the probability $p_i$; line 7, the step size $\eta^t = 1$.

**Theorem 4.4.** *Consider the Algorithm 1 instantiated with FedProx. If $\mu$ is chosen to satisfy $\mu > \lambda_{\min}$, and $K, \mu, \gamma$ is chosen properly such that $\kappa = \frac{1-\gamma B}{\mu} - \frac{LB(1+\gamma)}{\mu\bar{\mu}} - \frac{L(1+\gamma)^2 B^2}{2\bar{\mu}^2} - \frac{LB^2(1+\gamma)^2}{K\bar{\mu}^2}(2\sqrt{K}+1) - \frac{B(1+\gamma)}{\bar{\mu}\sqrt{K}} - \frac{LNB^2(1+\gamma)^2}{8K^2 p_{\min}\bar{\mu}^2} > 0$, then after $T$ rounds,*

$$\min_{t\in[T-1]}\mathbb{E}[\|\nabla f(w^t)\|] \leq \frac{f(w^0) - f(w^*)}{\kappa T}.$$

**Remark**

- The rate of convergence of FEDDISCRETE is the same as that of FedProx but with worse constant due to the discretization mechanism.

- One can also prove the convergence of the FEDDISCRETE when the local clients perform the fixed number of stochastic gradient descent steps as FedAVG. Then the convergence result is a special case of the FedPAQ (Reisizadeh et al., 2020) when $q$ is set to $1/4$.

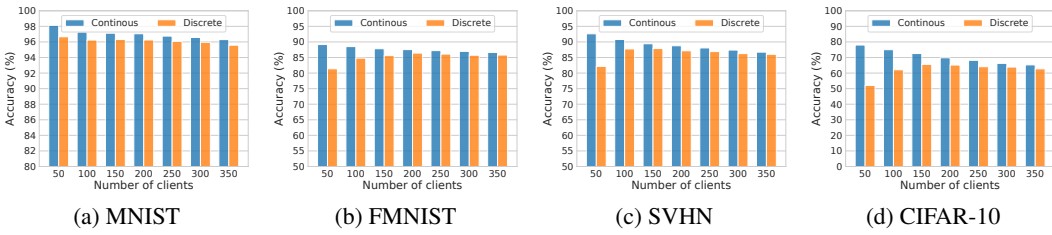

Figure 2: Effect of number of clients $N$ with i.i.d setting.

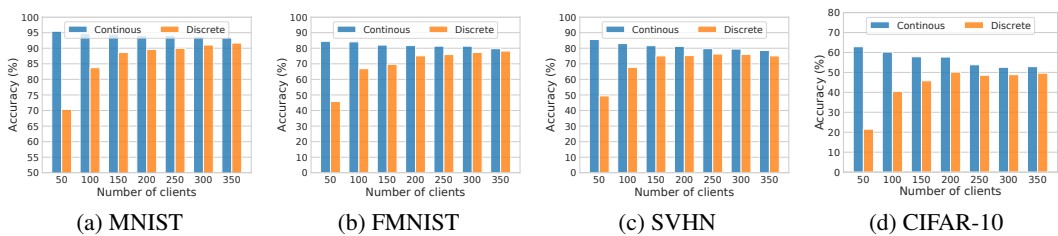

Figure 3: Effect of number of clients $N$ with non-i.i.d setting.

## 5 EXPERIMENTAL EVALUATION

In this section, we examine the effect of FEDDISCRETE on the four image benchmark datasets: MNIST (LeCun et al., 1998), Fashion-MNIST (FMNIST) (Xiao et al., 2017), SVHN (Netzer et al., 2011), and CIFAR-10 (Krizhevsky et al., 2009). For MNIST and FMNIST, we adopt a two-layer CNN for image classification; for SVHN and CIFAR-10, the small network from the Pytorch library[2] only achieves around 50% accuracy, so that we re-design a small VGG (Simonyan & Zisserman, 2014) for them. The training data and the testing data are fed into the network directly in each client, and for each client, the size of the training data is the total number of the training samples divided by the number of the clients for i.i.d setting. For non-i.i.d setting, we sort the data by digit labels first, then divide the datasets into $2N$ shards of size $\frac{|D|}{2N}$, and assign each of $N$ clients 2 shards, which is the same setting as (McMahan et al., 2017). We use the local SGD as the optimization algorithm for each client, where the local learning rate is set as $0.03$ for MNIST/FMNIST and $0.015$ for SVHN/CIFAR-10. And the global learning rate $\eta_t$ is set to $1$ for all $t$. Considering the randomness from the discrete mechanism, we run the test experiments five times independently to obtain an averaged value. To evaluate the performance of different approaches, we use different metrics including accuracy/utility for model performance, attack success rate (ASR)[3] for attack performance and the number of communication rounds (CRs), for communication cost. Any approach with a high accuracy and a low ASR indicates a good and practical defense solution. The proposed models are implemented using Pytorch, and all experiments are done with a single GPU NVIDIA Tesla V100 on the local server. Experiments on MNIST and FMNIST can be finished within an hour with 10 CRs, and experiments on SVHN and CIFAR-10 need about 2 hours with 15 CRs.

**Evaluation on Discrete Mechanism.** Here, we first test the effectiveness of the discrete mechanism in FL. In Figure 2 and 3, the results demonstrate three observations: 1) for the discrete model, in i.i.d setting, it performs closer to continuous model aggregation than that in the non-i.i.d setting; 2) the more complex and difficult tasks, i.e., CIFAR-10 > SVHN > FMNIST > MNIST, are more sensitive to the discrete mechanism; 3) when increasing the number of total clients, the model utility loss between continuous and discrete models are diminishing. For example, while having 350 clients in FL, the difference between continuous and discrete models is less than 1% and 2% for all tasks in the i.i.d and non-i.i.d settings, respectively. These results verify the previous analysis in Theorem

---

[2]https://pytorch.org/tutorials/beginner/blitz/cifar10_tutorial.html. Note that, we use the default setting provided from Pytorch for MNIST and Fashion-MNIST. For SVHN and CIFAR-10, we use VGG9 that is modified from VGG11 for balancing the memory usage and the model utility.

[3]The precise definition of ASR varies under different attacks. We will give the precise definition in the subsequent sections.

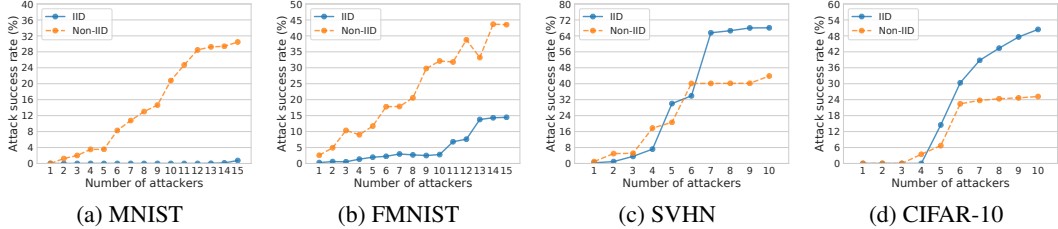

Figure 4: Effect of number of attackers $F$ with 100 clients in i.i.d and non-i.i.d settings.

4.3. In summary, the discrete mechanism is more practical for customer-related applications, e.g., smartphones or IoT devices, since it usually involves millions of clients during training.

**Evaluation on Availability Poisoning Attacks.** The integrity poisoning attacks are evaluated from multi-angles. In Figure 4, we fix the number of clients as 100, and evaluate the increasing number of attackers in both i.i.d and non-i.i.d settings. The attack success rate (ASR) in APAs is the utility loss between before and after attacks. The results show that one or a few attackers, i.e., a small fraction of attackers of all clients, cannot successfully poison the model. For example, when there are more than 100 clients, any single adversarial client only gets the 0% ASR for all tasks in both i.i.d and non-i.i.d settings. Figure 5 and 6 show the results of model utility with a fixed number of attackers in both i.i.d and non-i.i.d settings. The results consistently show that for a fixed number of attackers, increasing the number of the total clients will improve the global model's utility.

Compared with prior defense against IBAs (Blanchard et al., 2017; Bagdasaryan et al., 2020; Shen et al., 2016; Fung et al., 2018), based on our current knowledge, not many existing works have well studied the availability poisoning attacks in FL. Due to the adversary have more flexible attack options of the training process in APAs, most APAs could easily break the defense solutions for IBAs.

Table 1: FEDDISCRETE against IBAs

| Attack | No Defense | | FEDDISCRETE | |
|---|---|---|---|---|
| | ASR | UL | ASR | UL |
| CAS | 81.90% | 2.40% | 0.00% | 0.00% |
| DBA | 81.30% | 4.70% | 0.00% | 0.00% |

Table 2: Compare with other defense baselines

**Evaluation on Integrity Backdoor Attacks.** Next, we evaluate the FEDDISCRETE against the state-of-the-art backdoor attacks, such as constrain-and-scale (CAS) (Bagdasaryan et al., 2020) and DBA (Xie et al., 2019) in Table 1, where utility loss (UL) is the difference of model utility between clean and poisoned models, and the attack success rate (ASR) in IBAs is the percentage of poisoned samples that are classified as the target class by the backdoored model.

| Defenses | ASR | UL |
|---|---|---|
| No defense | 81.90% | 2.40% |
| Krum | 100% | 35.50% |
| FoolsGold | 100% | 39.90% |
| Auror | 100% | 66.10% |
| FedDP | 0% | 13.30% |
| FEDDISCRETE | 0% | 0% |

Due to the variants of studied tasks in prior works, we evaluate the most common task, i.e., CIFAR-10, for all backdoor attacks in the i.i.d setting. We use the default attack settings same as the original paper: one attacker for CAS and four attackers for DBA. Here, ASR is the backdoor attack success rate, and UL is the utility loss compared with none-attack FL. The results show that FEDDISCRETE can easily defend the backdoor attacks against CAS and DBA. By mitigating the impacts of the attackers, in addition to the successful IBAs defense, FEDDIS-CRETE can even improve the model utility.

We also compare FEDDISCRETE with other backdoor defense systems against CAS in FL: Krum (Blanchard et al., 2017), FoolsGold (Fung et al., 2018), Auror (Shen et al., 2016), and FedDP (Geyer et al., 2017) in Table 2. The results show that Krum, FoolsGold, and Auror can not well defend the backdoor attacks in FL. DP can achieve a good backdoor defense, but scarifies the model utility. FEDDISCRETE can achieve the best ASR and UL in all defense methods.

**Discussion on Client Inference Attacks.** Prior inference attacks (Blanchard et al., 2017; Pyrgelis et al., 2017; Ganju et al., 2018) require to get the original or similar local client model, i.e., a continuous local model. Then, the malicious server could explore the privacy of the training data from its

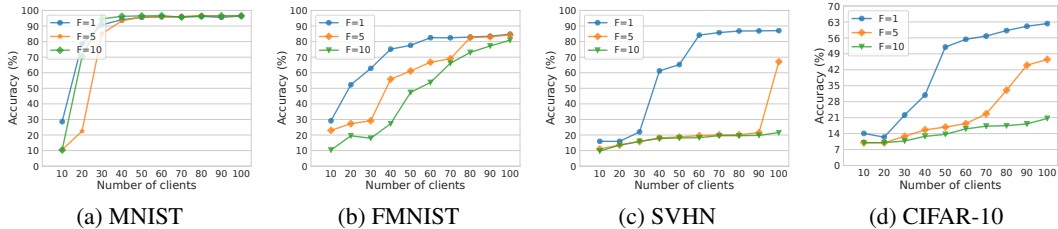

Figure 5: Effect of number of clients $N$ with fixed attackers ($F = 1, 5, 10$) in i.i.d settings.

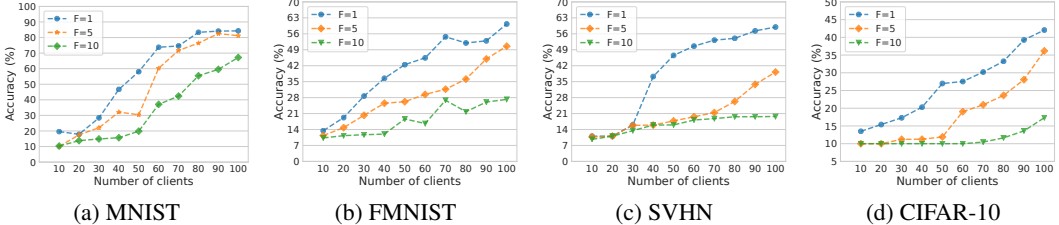

Figure 6: Effect of number of clients $N$ with fixed attackers ($F = 1, 5, 10$) in non-i.i.d settings.

continuous local model. However, a discrete local model cannot be used to infer any valuable information. The malicious server could compute the continuous global model, but it would not be useful to explore the privacy information of each client. Another popular inference defense technique is the homomorphic encryption (HE) (Gentry et al., 2009). Although HE can perfectly defend the inference attacks from the malicious server, it requires high computation cost and cannot defend against other weight-based poisoning attacks. Similarly, differential privacy (DP) (Geyer et al., 2017) could also be used for defending inference attacks. However, Hu et al. (2021) have shown that DP will lose too much utility of the global model while providing strong privacy protection. FEDDISCRETE is a concise but practical approach that can naturally defend the client inference attacks from the malicious server with a more efficient communication protocol.

## 6 LIMITATIONS AND DISCUSSIONS

FEDDISCRETE can defend both data poisoning attacks and weight poisoning attacks in FL, since data poisoning attacks are equal to change the clean model to the poisoned model. However, FED-DISCRETE also has two limitations: (1) It could not work in the cross-silos federated learning setting. If we want to mitigate the adversarial impact and achieve a non-biased aggregated model, it requires there are many clients could join in training in each round. However, it would not be an issue in the cross-devices setting involving thousands or even more clients in each round of training; (2) It could not defend the byzantine attacks through our discretization aggregation mechanism, since the aggregated model represents the majority of interests in FEDDISCRETE. However, the byzantine defense via robust local training could still work in our framework, where the local client can recognize the poisoned model or only leverage the valuable information from the poisoned model to train the local model via knowledge distillation (Lee et al., 2021). Our next step is to further explore a novel defense mechanism that can be more scalable for various adversarial environments.

## 7 CONCLUSION

In this paper, we proposed a new FL approach that applies the discrete mechanism with adaptive weight range for protecting FL. To our best knowledge, it is the first work that don't require the server to do excessive computation, but successfully defend against various attacks, including availability poisoning attacks, integrity backdoor attacks, and inference attacks. We also theoretically analyze the utility, robustness, and convergence of our proposed discrete mechanism in FL. One limitation of this work is the proposed system cannot defend the attacks well with a small number of clients or face a large fraction of the attackers, which becomes the next step of our future research.

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

# A APPENDIX

## A.1 PROOF OF LEMMA 4.2.

*Proof.* Notice that for any scalar $w \in \mathbb{R}^n$ and $l \leq w \leq u$, by the definition of $\mathcal{M}(.; l, u)$, one can verify that $\mathbb{E}[\mathcal{M}(w; l, u)] = w$, therefore $\mathbb{E}[\mathcal{M}(\boldsymbol{w}; l, u)] = \boldsymbol{w}$. Furthermore,

$$
\begin{aligned}
\mathbb{E}[\|\mathcal{M}(\boldsymbol{w}; l, u) - \boldsymbol{w}\|^2] &= \sum_{j=1}^{d} \mathbb{E}[([\mathcal{M}(\boldsymbol{w}; l, u)]_j - [\boldsymbol{w}]_j)^2] \\
&= \sum_{j=1}^{d} \left( \mathbb{E}[[\mathcal{M}(\boldsymbol{w}; l, u)]_j^2] - [\boldsymbol{w}]_j^2 \right) \\
&= \sum_{j=1}^{d} \left( (u+l)[\boldsymbol{w}]_j - lu - [\boldsymbol{w}]_j^2 \right) \\
&= \sum_{j=1}^{d} \left[ \left( \frac{u-l}{2} \right)^2 - \left( [\boldsymbol{w}]_j - \left( \frac{u+l}{2} \right) \right)^2 \right] \\
&= d\left( \frac{u-l}{2} \right)^2 - \sum_{j=1}^{d} \left( [\boldsymbol{w}]_j - \left( \frac{u+l}{2} \right) \right)^2 \leq \frac{d(u-l)^2}{4}.
\end{aligned}
$$

Moreover,

$$
d\left( \frac{u-l}{2} \right)^2 - \sum_{j=1}^{d} \left( [\boldsymbol{w}]_j - \left( \frac{u+l}{2} \right) \right)^2 =
$$

$\square$

## A.2 PROOF OF THEOREM 4.2.

Before proving the Theorem 4.2, a standard probability bound is required.

**Lemma A.1.** *Let $\{X_i\}_{i=1}^n$ be a sequence of i.i.d continuous random variables, whose support is on $\mathbb{R}$. Then for any $b \in \mathbb{R}$*

$$
\mathbb{P}\left( \max_{i \in [n]} X_i \geq b \right) \leq n\mathbb{P}(X_i \geq b).
$$

*Proof.*

$$
\begin{aligned}
\mathbb{P}\left( \max_{i \in [n]} X_i \geq b \right) &= 1 - \mathbb{P}(\max_{i \in [n]} X_i \leq b) = 1 - \mathbb{P}(X_1 \leq b, \cdots, X_n \leq b) \\
&= 1 - \prod_{j=1}^{n} \mathbb{P}(X_j \leq b) = 1 - (1 - \mathbb{P}(X_j \geq b))^n \text{ for any } j
\end{aligned}
$$

Let $f(t) := 1 - nt - (1-t)^n$ for $t \in [0, 1]$. As $f'(t) \leq 0$ and $f(0) = 0$, $f(t) \leq 0$ for all $t \in [0, 1]$. Therefore, $1 - (1-t)^n \leq nt$. Take $t = \mathbb{P}(X_j \geq b)$, we arrive

$$
\mathbb{P}\left( \max_{i \in [n]} X_i \geq b \right) \leq n\mathbb{P}(X_i \geq b).
$$

$\square$

Now we are ready to prove Theorem 4.2.

*Proof.* For any $j \in [d]$, $|[\bar{\boldsymbol{w}}^t]_j - [\bar{\boldsymbol{w}}^t_{\mathcal{M}}]_j| \leq u^t - l^t$ and $\mathbf{Var}\left([\bar{\boldsymbol{w}}^t]_j - [\bar{\boldsymbol{w}}^t_{\mathcal{M}}]_j\right) \leq (u^t - l^t)^2/4$ due to Lemma 4.1. By Bernstein inequality, for any $j \in [d]$ and $\epsilon > 0$

$$\mathbb{P}\left(\left|[\bar{\boldsymbol{w}}^t]_j - [\bar{\boldsymbol{w}}^t_{\mathcal{M}}]_j\right| \geq \epsilon\right) \leq 2\exp\left(-\frac{K\epsilon^2}{2\frac{1}{K}\sum_{i \in S_t}\mathbf{Var}\left([\bar{\boldsymbol{w}}^t]_j - [\bar{\boldsymbol{w}}^t_{\mathcal{M}}]_j\right) + \frac{2}{3}\epsilon(u^t - l^t)}\right)$$

$$\leq 2\exp\left(-\frac{K\epsilon^2}{\frac{(u^t - l^t)^2}{2} + \frac{2}{3}\epsilon(u^t - l^t)}\right)$$

By Lemma A.1,

$$\mathbb{P}\left(\max_{j \in [d]}\left|[\bar{\boldsymbol{w}}^t]_j - [\bar{\boldsymbol{w}}^t_{\mathcal{M}}]_j\right| \geq \epsilon\right) \leq 2d\exp\left(-\frac{K\epsilon^2}{\frac{(u^t - l^t)^2}{2} + \frac{2}{3}\epsilon(u^t - l^t)}\right)$$

Therefore, for any $\beta > 0$, there exists $\epsilon = \mathcal{O}\left(\frac{(u^t - l^t)\sqrt{\log\frac{2d}{\beta}}}{\sqrt{K}}\right)$ such that $\mathbb{P}\left(\max_{j \in [d]}|[\bar{\boldsymbol{w}}^t]_j - [\bar{\boldsymbol{w}}^t_{\mathcal{M}}]_j| \leq \epsilon\right)$ holds with probability at least $1 - \beta$. □

## A.3 PROOF OF THEOREM 4.3

*Proof.* Due to the weight discretization mechanism, the adversary can only return $u$ or $l$ for each coordinate of the model weight. In order to not return the correct information, the adversary could choose to return the opposite feedback to attack the model, i.e., return $u$ if the original return is $l$, and vice versa. Therefore, we denote, for any $l \leq w \leq u$,

$$\mathcal{M}_{\text{adv}}(w) = \begin{cases} l, & \text{w.p.}\frac{w-l}{h-l} \\ h, & \text{w.p.}\frac{h-w}{h-l} \end{cases}$$

Under the scenario that there are $F$ attackers, for any $j \in [d]$,

$$[\mathbb{E}[\bar{\boldsymbol{w}}_{\mathcal{M}}]]_j = \mathbb{E}\left[\frac{1}{N}\left(\sum_{i=1}^{N-F}[\mathcal{M}(\boldsymbol{w}_i)]_j + \sum_{i=N-F+1}^{N}[\mathcal{M}_{\text{adv}}(\boldsymbol{w}_i)]_j\right)\right]$$

$$= \frac{1}{N}\sum_{i=1}^{N-F}[\boldsymbol{w}_i]_j + \frac{1}{N}\sum_{i=N-F+1}^{N}((h+l) - [\boldsymbol{w}_i]_j) \tag{4}$$

$$= \frac{1}{N}\left(\sum_{i=1}^{N-F}[\boldsymbol{w}_i]_j - \sum_{i=N-F+1}^{N}[\boldsymbol{w}_i]_j\right) + (h+l)\frac{F}{N}. \tag{5}$$

□

## A.4 PROOF OF THEOREM 4.4

*Proof.* The proof is inspired by the analysis in Theorem 4 of (Li et al., 2020).

To proceed with the analysis, we first introduce some notations. At the $t$th round, for the all $i \in \mathcal{S}'_t$, define $\tilde{\boldsymbol{w}}^{t+1} = \boldsymbol{w}^t + \frac{1}{|\mathcal{S}'_t|}\sum_{i \in \mathcal{S}'_t}\left(\boldsymbol{w}_i^{t+1} - \boldsymbol{w}^t\right)$, $\bar{\boldsymbol{w}}^{t+1} = \boldsymbol{w}^t + \sum_{i \in [N]}p_i(\boldsymbol{w}_i^{t+1} - \boldsymbol{w}^t)$, and $\hat{\boldsymbol{w}}_i^{t+1} = \arg\min_{\boldsymbol{w}} h_i(\boldsymbol{w}; \boldsymbol{w}^t) := F_i(\boldsymbol{w}) + \frac{\mu}{2}\|\boldsymbol{w} - \boldsymbol{w}^t\|^2$. $\tilde{\boldsymbol{w}}^{t+1}$ is the ghost global model as if the discretization mechanism is not applied to the local model weights; $\bar{\boldsymbol{w}}^{t+1}$ is another ghost global model as if all clients participate in the $t$th round training and no discretization mechanism is applied ; $\hat{\boldsymbol{w}}_i^{t+1}$ is the exact minimizer of the strongly convex function $h_i(\boldsymbol{w})$. These points reference points are crucial for the analysis. Define the gradient residual $e_i^{t+1} = \nabla F_i(\boldsymbol{w}_i^{t+1}) + \mu(\boldsymbol{w}_i^{t+1} - \boldsymbol{w}^t)$, then $\boldsymbol{w}_i^{t+1} - \boldsymbol{w}^t = -\frac{1}{\mu}\nabla F_i(\boldsymbol{w}_i^{t+1}) + \frac{1}{\mu}e_i^{t+1}$. Therefore,

$$\bar{\boldsymbol{w}}^{t+1} - \boldsymbol{w}^t = \sum_{i \in [n]}p_i(\boldsymbol{w}_i^{t+1} - \boldsymbol{w}^t) = -\frac{1}{\mu}\sum_{i \in [n]}p_i\nabla F_i(\boldsymbol{w}_i^{t+1}) + \frac{1}{\mu}\sum_{i \in [n]}p_ie_i^{t+1}. \tag{6}$$

Since $\mu$ is chosen to satisfy $\mu > \lambda_{\min}$, then $h_i(\boldsymbol{w}; \boldsymbol{w}^t)$ is $\bar{\mu}$-strongly convex. By the strong convexity of $h_i$,

$$\left\| \boldsymbol{w}_i^{t+1} - \hat{\boldsymbol{w}}_i^{t+1} \right\|^2 \leq \frac{1}{\bar{\mu}} (\boldsymbol{w}_i^{t+1} - \hat{\boldsymbol{w}}_i^{t+1})^\top (\nabla h_i(\boldsymbol{w}_i^{t+1}) - \nabla h_i(\hat{\boldsymbol{w}}_i^{t+1}))$$

$$\leq \frac{1}{\bar{\mu}} \left\| \boldsymbol{w}_i^{k+1} - \hat{\boldsymbol{w}}_i^{t+1} \right\| \left\| \nabla h_i(\boldsymbol{w}_i^{t+1}) - \nabla h_i(\hat{\boldsymbol{w}}_i^{t+1}) \right\|,$$

which, together with the fact that $\hat{\boldsymbol{w}}_i^{t+1}$ is the minimizer of $h_i(\boldsymbol{w})$, implies

$$\left\| \boldsymbol{w}_i^{t+1} - \hat{\boldsymbol{w}}_i^{t+1} \right\| \leq \frac{1}{\bar{\mu}} \left\| \nabla h_i(\boldsymbol{w}_i^{t+1}) - \nabla h_i(\hat{\boldsymbol{w}}_i^{t+1})) \right\|$$

$$= \frac{1}{\bar{\mu}} \left\| \nabla h_i(\hat{\boldsymbol{w}}_i^{t+1})) \right\| = \frac{1}{\bar{\mu}} \left\| \nabla F_i(\boldsymbol{w}_i^{t+1}) + \mu(\boldsymbol{w}_i^{t+1} - \boldsymbol{w}^t) \right\|$$

$$\leq \frac{\gamma}{\bar{\mu}} \left\| \nabla F_i(\boldsymbol{w}^t) \right\| \tag{7}$$

Again use the same analysis, one has $\left\| \hat{\boldsymbol{w}}_i^{t+1} - \boldsymbol{w}^t \right\| \leq \frac{1}{\bar{\mu}} \nabla F_i(\boldsymbol{w}^t)$. Therefore, together with Eq. 7,

$$\left\| \boldsymbol{w}_i^{t+1} - \boldsymbol{w}^t \right\| \leq \left\| \boldsymbol{w}_i^{t+1} - \hat{\boldsymbol{w}}_i^{t+1} \right\| + \left\| \hat{\boldsymbol{w}}_i^{t+1} - \boldsymbol{w}^t \right\| \leq \frac{1+\gamma}{\bar{\mu}} \left\| \nabla F_i(\boldsymbol{w}^t) \right\|. \tag{8}$$

Therefore, one can bound the distance from the ghost global model $\bar{\boldsymbol{w}}^{t+1}$ to the current global weight as

$$\left\| \bar{\boldsymbol{w}}^{t+1} - \boldsymbol{w}^t \right\| = \left\| \sum_{i \in [N]} p_i (\boldsymbol{w}_i^{t+1} - \boldsymbol{w}^t) \right\| \leq \sum_{i \in [N]} p_i \left\| \boldsymbol{w}_i^{t+1} - \boldsymbol{w}^t \right\|$$

$$\leq \frac{1+\gamma}{\bar{\mu}} \sum_{i \in [N]} p_i \left\| \nabla F_i(\boldsymbol{w}^t) \right\| \qquad \text{(by Eq. 8)}$$

$$\leq \frac{1+\gamma}{\bar{\mu}} \sqrt{\sum_{i \in [N]} p_i \left\| \nabla F_i(\boldsymbol{w}^t) \right\|^2} \qquad \text{(Jensen' Inequality)}$$

$$= \frac{1+\gamma}{\bar{\mu}} \sqrt{\mathbb{E}_i[\left\| \nabla F_i(\boldsymbol{w}^t) \right\|^2]}$$

$$\leq \frac{B(1+\gamma)}{\bar{\mu}} \left\| \nabla f(\boldsymbol{w}^t) \right\| \qquad \text{(by Assumption 4.1 (3))} \tag{9}$$

Note that

$$\left\| \sum_{i \in [n]} p_i \left( \nabla F_i(\boldsymbol{w}_i^{t+1}) - e_i^{t+1} - \nabla F_i(\boldsymbol{w}^t) \right) \right\| \leq \sum_{i \in [n]} p_i \left( \left\| \nabla F_i(\boldsymbol{w}_i^{t+1}) - \nabla F_i(\boldsymbol{w}^t) \right\| + \left\| e_i^{t+1} \right\| \right)$$

$$\leq \sum_{i \in [n]} p_i \left( L \left\| \boldsymbol{w}_i^{t+1} - \boldsymbol{w}^t \right\| + \left\| e_i^{t+1} \right\| \right)$$

$$\overset{Eq.\ 8}{\leq} \left( \frac{L(1+\gamma)}{\bar{\mu}} + \gamma \right) \sum_{i \in [n]} p_i \left\| \nabla F_i(\boldsymbol{w}^t) \right\|$$

$$= \left( \frac{L(1+\gamma)}{\bar{\mu}} + \gamma \right) \mathbb{E}_i[\nabla F_i(\boldsymbol{w}^t)]$$

$$\overset{Eq.\ 8}{\leq} B \left( \frac{L(1+\gamma)}{\bar{\mu}} + \gamma \right) \left\| \nabla f(\boldsymbol{w}^t) \right\|. \tag{10}$$

By Assumption 4.1 (1), one has

$$f(\bar{\boldsymbol{w}}^{t+1}) \leq f(\boldsymbol{w}^t) + \nabla f(\boldsymbol{w}^t)^\top (\bar{\boldsymbol{w}}^{t+1} - \boldsymbol{w}^t) + \frac{L}{2} \left\| \bar{\boldsymbol{w}}^{t+1} - \boldsymbol{w}^t \right\|^2$$

$$\overset{Eq.\ 6}{\leq} f(\boldsymbol{w}^t) + \nabla f(\boldsymbol{w}^t)^\top \left( -\frac{1}{\mu} \sum_{i \in [n]} p_i \nabla F_i(\boldsymbol{w}_i^{t+1}) + \frac{1}{\mu} \sum_{i \in [n]} p_i e_i^{t+1} \right) + \frac{L}{2} \left\| \bar{\boldsymbol{w}}^{t+1} - \boldsymbol{w}^t \right\|^2$$

$$= f(\boldsymbol{w}^t) + \nabla f(\boldsymbol{w}^t)^\top \left( -\frac{1}{\mu} \sum_{i \in [n]} p_i \left( \nabla F_i(\boldsymbol{w}_i^{t+1}) - e_i^{t+1} - \nabla F_i(\boldsymbol{w}^t) \right) - \frac{1}{\mu} \nabla f(\boldsymbol{w}^t) \right)$$
$$+ \frac{L}{2} \left\| \bar{\boldsymbol{w}}^{t+1} - \boldsymbol{w}^t \right\|^2$$

$$\leq f(\boldsymbol{w}^t) - \frac{1}{\mu} \left\| f(\boldsymbol{w}^t) \right\|^2 - \frac{1}{\mu} f(\boldsymbol{w}^t)^\top \left( \sum_{i \in [n]} p_i \left( \nabla F_i(\boldsymbol{w}_i^{t+1}) - e_i^{t+1} - \nabla F_i(\boldsymbol{w}^t) \right) \right)$$
$$+ \frac{L}{2} \left\| \bar{\boldsymbol{w}}^{t+1} - \boldsymbol{w}^t \right\|^2$$

$$\leq f(\boldsymbol{w}^t) - \frac{1}{\mu} \left\| f(\boldsymbol{w}^t) \right\|^2 + \frac{1}{\mu} \left\| f(\boldsymbol{w}^t) \right\| \left\| \sum_{i \in [n]} p_i \left( \nabla F_i(\boldsymbol{w}_i^{t+1}) - e_i^{t+1} - \nabla F_i(\boldsymbol{w}^t) \right) \right\|$$
$$+ \frac{L}{2} \left\| \bar{\boldsymbol{w}}^{t+1} - \boldsymbol{w}^t \right\|^2$$

$$\overset{Eq.\ 10, Eq.\ 9}{\leq} f(\boldsymbol{w}^t) - \frac{1}{\mu} \left\| f(\boldsymbol{w}^t) \right\|^2 + \frac{B}{\mu} \left( \frac{L(1+\gamma)}{\bar{\mu}} + \gamma \right) \left\| \nabla f(\boldsymbol{w}^t) \right\|^2 + \frac{L}{2} (\frac{B(1+\gamma)}{\bar{\mu}})2 \left\| \nabla f(\boldsymbol{w}^t) \right\|^2 \tag{11}$$

$$= f(\boldsymbol{w}^t) - \left( \frac{1 - \gamma B}{\mu} - \frac{LB(1+\gamma)}{\mu\bar{\mu}} - \frac{L(1+\gamma)^2 B^2}{2\bar{\mu}^2} \right) \left\| \nabla f(\boldsymbol{w}^t) \right\|^2 \tag{12}$$

By mean-value theorem and triangular inequality, for some $\alpha \in [0, 1]$

$$\begin{aligned}
f(\tilde{\boldsymbol{w}}^{t+1}) &\leq f(\bar{\boldsymbol{w}}^{t+1}) + \left\| \nabla f(\alpha \tilde{\boldsymbol{w}}^{t+1} + (1-\alpha)\bar{\boldsymbol{w}}^{t+1}) \right\| \left\| \tilde{\boldsymbol{w}}^{t+1} - \bar{\boldsymbol{w}}^{t+1} \right\| \\
&\leq f(\bar{\boldsymbol{w}}^{t+1}) + \left( \left\| \nabla f(\alpha \tilde{\boldsymbol{w}}^{t+1} + (1-\alpha)\bar{\boldsymbol{w}}^{t+1}) - \nabla f(\boldsymbol{w}^t) \right\| + \left\| \nabla f(\boldsymbol{w}^t) \right\| \right) \left\| \tilde{\boldsymbol{w}}^{t+1} - \bar{\boldsymbol{w}}^{t+1} \right\| \\
&\leq f(\bar{\boldsymbol{w}}^{t+1}) + \left( L \left\| \alpha \tilde{\boldsymbol{w}}^{t+1} + (1-\alpha)\bar{\boldsymbol{w}}^{t+1} - \boldsymbol{w}^t \right\| + \left\| \nabla f(\boldsymbol{w}^t) \right\| \right) \left\| \tilde{\boldsymbol{w}}^{t+1} - \bar{\boldsymbol{w}}^{t+1} \right\| \\
&\leq f(\bar{\boldsymbol{w}}^{t+1}) + \left( L(\left\| \tilde{\boldsymbol{w}}^{t+1} - \boldsymbol{w}^t \right\| + \left\| \bar{\boldsymbol{w}}^{t+1} - \boldsymbol{w}^t \right\|) + \left\| \nabla f(\boldsymbol{w}^t) \right\| \right) \left\| \tilde{\boldsymbol{w}}^{t+1} - \bar{\boldsymbol{w}}^{t+1} \right\|
\end{aligned} \tag{13}$$

Taking expectation with respect to the random index set $\mathcal{S}_t'$, one gets

$$\begin{aligned}
\mathbb{E}_{\mathcal{S}_t'}[f(\tilde{\boldsymbol{w}}^{t+1})] &\leq f(\bar{\boldsymbol{w}}^{t+1}) + \left( L \left\| \bar{\boldsymbol{w}}^{t+1} - \boldsymbol{w}^t \right\| + \left\| \nabla f(\boldsymbol{w}^t) \right\| \right) \mathbb{E}_{\mathcal{S}_t'} \left\| \tilde{\boldsymbol{w}}^{t+1} - \bar{\boldsymbol{w}}^{t+1} \right\| \\
&\quad + L \mathbb{E}_{\mathcal{S}_t'}[\left\| \tilde{\boldsymbol{w}}^{t+1} - \boldsymbol{w}^t \right\| \left\| \tilde{\boldsymbol{w}}^{t+1} - \bar{\boldsymbol{w}}^{t+1} \right\|] \\
&\leq f(\bar{\boldsymbol{w}}^{t+1}) + \left( L \left\| \bar{\boldsymbol{w}}^{t+1} - \boldsymbol{w}^t \right\| + \left\| \nabla f(\boldsymbol{w}^t) \right\| \right) \mathbb{E}_{\mathcal{S}_t'} \left\| \tilde{\boldsymbol{w}}^{t+1} - \bar{\boldsymbol{w}}^{t+1} \right\| \\
&\quad + L \mathbb{E}_{\mathcal{S}_t'}[(\left\| \tilde{\boldsymbol{w}}^{t+1} - \bar{\boldsymbol{w}}^{t+1} \right\| + \left\| \bar{\boldsymbol{w}}^{t+1} - \boldsymbol{w}^t \right\|) \left\| \tilde{\boldsymbol{w}}^{t+1} - \bar{\boldsymbol{w}}^{t+1} \right\|] \\
&= f(\bar{\boldsymbol{w}}^{t+1}) + \left( 2L \left\| \bar{\boldsymbol{w}}^{t+1} - \boldsymbol{w}^t \right\| + \left\| \nabla f(\boldsymbol{w}^t) \right\| \right) \mathbb{E}_{\mathcal{S}_t'} \left\| \tilde{\boldsymbol{w}}^{t+1} - \bar{\boldsymbol{w}}^{t+1} \right\| \\
&\quad + L \mathbb{E}_{\mathcal{S}_t'}[\left\| \tilde{\boldsymbol{w}}^{t+1} - \bar{\boldsymbol{w}}^{t+1} \right\|^2]
\end{aligned} \tag{14}$$

By the sampling scheme, one has

$$
\begin{aligned}
\mathbb{E}_{\mathcal{S}'_t} \left\| \tilde{\boldsymbol{w}}^{t+1} - \bar{\boldsymbol{w}}^{t+1} \right\|^2 &\leq \frac{1}{|\mathcal{S}'_t|} \mathbb{E}_i[\left\| \boldsymbol{w}_i^{t+1} - \bar{\boldsymbol{w}}^{t+1} \right\|^2] \\
&\leq \frac{1}{|\mathcal{S}'_t|} \mathbb{E}_i[\left\| \boldsymbol{w}_i^{t+1} - \boldsymbol{w}^t \right\|^2 + \left\| \boldsymbol{w}^t - \bar{\boldsymbol{w}}^{t+1} \right\|^2 + 2(\boldsymbol{w}_i^{t+1} - \boldsymbol{w}^t)^\top (\boldsymbol{w}^t - \bar{\boldsymbol{w}}^{t+1})] \\
&\leq \frac{1}{|\mathcal{S}'_t|} \mathbb{E}_i[\left\| \boldsymbol{w}_i^{t+1} - \boldsymbol{w}^t \right\|^2] \qquad (\text{by } \mathbb{E}_i(\boldsymbol{w}_i^{t+1}) = \bar{\boldsymbol{w}}^{t+1}) \\
&\overset{Eq.\ 8}{\leq} \frac{1}{|\mathcal{S}'_t|} \frac{(1+\gamma)^2}{\bar{\mu}^2} \mathbb{E}_i[\left\| \nabla F_i(\boldsymbol{w}^t) \right\|^2] \\
&\leq \frac{B^2}{|\mathcal{S}'_t|} \frac{(1+\gamma)^2}{\bar{\mu}^2} \left\| \nabla f(\boldsymbol{w}^t) \right\|^2 \quad (\text{by Assumption } 4.1\ (3)).
\end{aligned}
\tag{15}
$$

Combining Eq. 14, Eq. 15, and Eq. 9, together with the fact that $\mathbb{E}_{\mathcal{S}'_t} \left\| \tilde{\boldsymbol{w}}^{t+1} - \bar{\boldsymbol{w}}^{t+1} \right\| \leq \sqrt{\mathbb{E}_{\mathcal{S}'_t} \left\| \tilde{\boldsymbol{w}}^{t+1} - \bar{\boldsymbol{w}}^{t+1} \right\|^2}$ as a result of the Jesen's inequality, one reaches to

$$
\begin{aligned}
\mathbb{E}_{\mathcal{S}'_t}[f(\tilde{\boldsymbol{w}}^{t+1})] &\leq f(\bar{\boldsymbol{w}}^{t+1}) + \left( 2L \frac{B^2}{\sqrt{|\mathcal{S}'_t|}} \frac{(1+\gamma)^2}{\bar{\mu}^2} + \frac{B}{\sqrt{|\mathcal{S}'_t|}} \frac{(1+\gamma)}{\bar{\mu}} + L \frac{B^2}{|\mathcal{S}'_t|} \frac{(1+\gamma)^2}{\bar{\mu}^2} \right) \left\| \nabla f(\boldsymbol{w}^t) \right\|^2 \\
&= f(\bar{\boldsymbol{w}}^{t+1}) + \left( \frac{LB^2(1+\gamma)^2}{|\mathcal{S}'_t|\bar{\mu}^2} (2\sqrt{|\mathcal{S}'_t|} + 1) + \frac{B(1+\gamma)}{\bar{\mu}\sqrt{|\mathcal{S}'_t|}} \right) \left\| \nabla f(\boldsymbol{w}^t) \right\|^2.
\end{aligned}
\tag{16}
$$

Combine Eq. 16 and Eq. 11, one reaches to

$$
\begin{aligned}
\mathbb{E}_{\mathcal{S}'_t}[f(\tilde{\boldsymbol{w}}^{t+1})] \leq f(\boldsymbol{w}^t) - \bigg( &\frac{1 - \gamma B}{\mu} - \frac{LB(1+\gamma)}{\mu\bar{\mu}} - \frac{L(1+\gamma)^2 B^2}{2\bar{\mu}^2} - \\
&\left( \frac{LB^2(1+\gamma)^2}{|\mathcal{S}'_t|\bar{\mu}^2} (2\sqrt{|\mathcal{S}'_t|} + 1) + \frac{B(1+\gamma)}{\bar{\mu}\sqrt{|\mathcal{S}'_t|}} \right) \bigg) \left\| \nabla f(\boldsymbol{w}^t) \right\|^2
\end{aligned}
\tag{17}
$$

Taking the expectation with respect to the discretization mechanism, $\mathbb{E}_{\mathcal{M}}[\boldsymbol{w}^{t+1}] = \tilde{\boldsymbol{w}}^{t+1}$ by Lemma 4.1. Since $f$ is $L$-smooth,

$$\mathbb{E}_{\mathcal{M}}[f(\boldsymbol{w}^{t+1})] \leq f(\tilde{\boldsymbol{w}}^{t+1}) + \frac{L}{2}\mathbb{E}_{\mathcal{M}}[\|\boldsymbol{w}^{t+1} - \tilde{\boldsymbol{w}}^{t+1}\|^2]$$

$$= f(\tilde{\boldsymbol{w}}^{t+1}) + \frac{L}{2}\frac{1}{|\mathcal{S}'_t|^2}\mathbb{E}_{\mathcal{M}}\left[\left\|\sum_{i\in\mathcal{S}'_t}\left(\mathcal{M}(\boldsymbol{w}_i^{t+1} - \boldsymbol{w}) - (\boldsymbol{w}_i^{t+1} - \boldsymbol{w})\right)\right\|^2\right]$$

$$\leq f(\tilde{\boldsymbol{w}}^{t+1}) + \frac{L}{8|\mathcal{S}'_t|^2}\left\|\sum_{i\in\mathcal{S}'_t}(\boldsymbol{w}_i^{t+1} - \boldsymbol{w})\right\|^2 \qquad \text{(by Lemma 4.1)}$$

$$\leq f(\tilde{\boldsymbol{w}}^{t+1}) + \frac{L}{8|\mathcal{S}'_t|^2}\left(\sum_{i\in[n]}\|\boldsymbol{w}_i^{t+1} - \boldsymbol{w}\|^2 + \sum_{i\neq j}(\boldsymbol{w}_i^{t+1} - \boldsymbol{w})^\top(\boldsymbol{w}_j^{t+1} - \boldsymbol{w})\right)$$

$$\leq f(\tilde{\boldsymbol{w}}^{t+1}) + \frac{L}{8|\mathcal{S}'_t|^2}\left(\sum_{i\in[n]}\|\boldsymbol{w}_i^{t+1} - \boldsymbol{w}\|^2 + \sum_{i\neq j}\|\boldsymbol{w}_i^{t+1} - \boldsymbol{w}\|\|\boldsymbol{w}_j^{t+1} - \boldsymbol{w}\|\right)$$

$$\leq f(\tilde{\boldsymbol{w}}^{t+1}) + \frac{LN}{8|\mathcal{S}'_t|^2}\left(\sum_{i\in[n]}\|\boldsymbol{w}_i^{t+1} - \boldsymbol{w}\|^2\right) \qquad \text{(by } 2\|a\|\|b\| \leq \|a\|^2 + \|b\|^2\text{)}$$

$$\leq f(\tilde{\boldsymbol{w}}^{t+1}) + \frac{LN}{8|\mathcal{S}'_t|^2 p_{\min}}\left(\sum_{i\in[n]}p_{\min}\|\boldsymbol{w}_i^{t+1} - \boldsymbol{w}\|^2\right)$$

$$\leq f(\tilde{\boldsymbol{w}}^{t+1}) + \frac{LN}{8|\mathcal{S}'_t|^2 p_{\min}}\left(\sum_{i\in[n]}p_i\|\boldsymbol{w}_i^{t+1} - \boldsymbol{w}\|^2\right)$$

$$\leq f(\tilde{\boldsymbol{w}}^{t+1}) + \frac{LN}{8|\mathcal{S}'_t|^2 p_{\min}}\left(\sum_{i\in[n]}p_i\frac{(1+\gamma)^2}{\bar{\mu}^2}\|\nabla F_i(\boldsymbol{w}^t)\|^2\right)$$

$$\leq f(\tilde{\boldsymbol{w}}^{t+1}) + \frac{LN}{8|\mathcal{S}'_t|^2 p_{\min}}\frac{B^2(1+\gamma)^2}{\bar{\mu}^2}\|\nabla f(\boldsymbol{w}^t)\|^2 \qquad (18)$$

Put Eq. 18 and Eq. 17 together, we reach to

$$\mathbb{E}_{\mathcal{M},\mathcal{S}'_t}[f(\boldsymbol{w}^{t+1})] \leq f(\boldsymbol{w}^t) - \left(\frac{1-\gamma B}{\mu} - \frac{LB(1+\gamma)}{\mu\bar{\mu}} - \frac{L(1+\gamma)^2B^2}{2\bar{\mu}^2} - \right.$$
$$\left.\left(\frac{LB^2(1+\gamma)^2}{|\mathcal{S}'_t|\bar{\mu}^2}(2\sqrt{|\mathcal{S}'_t|} + 1) + \frac{B(1+\gamma)}{\bar{\mu}\sqrt{|\mathcal{S}'_t|}}\right) - \frac{LNB^2(1+\gamma)^2}{8|\mathcal{S}'_t|^2 p_{\min}\bar{\mu}^2}\right)\|\nabla f(\boldsymbol{w}^t)\|^2$$
$$(19)$$

When there is no adversary, then $|\mathcal{S}'| = K$, then

$$\mathbb{E}_{\mathcal{M},\mathcal{S}'_t}[f(\boldsymbol{w}^{t+1})] \leq f(\boldsymbol{w}^t) - \kappa\|\nabla f(\boldsymbol{w}^t)\|^2$$

Finally, taking the total expectation with respect to all randomness and by telescoping, one reaches

$$\kappa\sum_{t=0}^{T-1}\mathbb{E}[\|\nabla f(\boldsymbol{w}_t)\|]^2 \leq f(\boldsymbol{w}_0) - f(\boldsymbol{w}^*).$$

Divide $T$ on both sides, then

$$\min_{t\in[T-1]}\mathbb{E}[\|\nabla f(\boldsymbol{w}_t)\|] \leq \frac{1}{T}\sum_{t=0}^{T-1}\mathbb{E}[\|\nabla f(\boldsymbol{w}_t)\|]^2 \leq \frac{f(\boldsymbol{w}_0) - f(\boldsymbol{w}^*)}{\kappa T}.$$

$$\square$$

