# OpenReview forum: "FedDiscrete: A Secure Federated Learning Algorithm Against Weight Poisoning"
_ICLR.cc/2022/Conference — ICLR 2022 Submitted_

### Official Review · Reviewer_kHbi · 2021-10-17

**Correctness:** 2
**Technical Novelty And Significance:** 1
**Empirical Novelty And Significance:** 1
**Recommendation:** 3
**Confidence:** 4

**Main Review:**

The work lacks novelty. Stochastic quantization is already well-studied in federated learning [1]. Binary quantization is just a special case of more general k-level quantization. The robustness result and the remark do not make sense to me. The remark seems to say that if poisoning rate and $|u-l|$ is small, the approach is robust. However, when these constraints hold, simple clipping would have the same effect so why bother to discretize?

The writing also needs huge improvement.

Overall, I do not recommend accepting the paper.

[1] McMahan, Brendan, Eider Moore, Daniel Ramage, Seth Hampson, and Blaise Aguera y Arcas. "Communication-efficient learning of deep networks from decentralized data." In Artificial intelligence and statistics, pp. 1273-1282. PMLR, 2017.

**Summary Of The Paper:**

The paper proposes to use stochastic quantization to increase robustness of federated learning.

**Summary Of The Review:**

I do not recommend acceptance because of (1) lack of novelty; (2) writing quality; (3) robustness results are hard to interpret.

---

### Official Review · Reviewer_zPLT · 2021-11-02

**Correctness:** 3
**Technical Novelty And Significance:** 3
**Empirical Novelty And Significance:** 3
**Recommendation:** 5
**Confidence:** 3

**Main Review:**

The main contribution of this work is the use of discretization mechanism for better robustness of federated learning, which is novel as far as I see. It is interesting that the mechanism improves robustness and privacy while sacrificing little utility when the ratio of adversaries is small. The proofs in the appendix are not carefully checked.

My main comments follow.
1.	The paper is not well-written. For example, the proof of Lemma 4.1 is called Lemma 4.2 in the appendix, and it is not completed. Moreover, some sentences are difficult to understand, e.g., in the introduction of the availability poisoning attacks (APA) in section 2, “the attacker can either control the global model’s utility on target tasks or decrease the most of the model’s utility as the optimal attack strategy.” Is difficult to understand and it does not introduce well on what APA does exactly. Also, in section 6 it says the proposed method could not defend Byzantine attacks, but it also says some Byzantine defense could still work with the proposed method. Can the authors elaborate more on this?
2.	The robust convergence guarantee may be vacuous. I’m worried that the assumption required for Theorem 4.4 can be easily invalid. In theorem 4.4, it requires the assumption of $\kappa>0$, where $\kappa$ is a sophisticate term defined by a term subtracting several positive terms. Where is the definition of $\bar{\mu}$ in the definition of $\kappa$? More importantly, when does this assumption $\kappa>0$ holds? Moreover, from the experiments it seems that the proposed method can only tolerate a small fraction of adversaries (e.g., up to 10%), which consolidates my worry that the robustness guarantee is vacuous.
3.	The experiments could be more complete. For example, I’m wondering how other baselines perform in the experiment of availability poisoning attack (Figure 4,5,6). Currently it only shows the proposed method. Moreover, I think many details of the experiments are not revealed. For example, how does the APA in your experiments performed?
4.	The security claim may not be well-supported. I understand the intuition that the discretization mechanism improves the privacy and security, but I’m not convinced that “a discrete local model cannot be used to infer any valuable information” as claimed in the paper. There are only general discussions about the security claim, but no supported theorems (like differential privacy guarantee) nor supported empirical evidence (client inference attack experiments).


Minor comments

In line 4 of Algorithm 1, are there typos in the definition of $l^t_i, u^t_i$? It reads as subtracting a vector from a scalar.


**Summary Of The Paper:**

This paper proposes a secure federated learning framework against weight poisoning. The key component in the framework is the discretization mechanism which works well with a sufficient number of clients. Theoretical analysis of the robustness and convergence is provided. Lastly, numerical analysis is provided to verify the performance of the proposed method.

**Summary Of The Review:**

The idea of discretization mechanism for the purpose of robustness is novel and interesting to me, but I’m worried about the quality about the paper: (1) the paper is not well-written; (2) the robust convergence guarantee may be vacuous; (3) the experiments could be more complete; (4) the security claim is not well-supported.

---

### Official Review · Reviewer_GZ4V · 2021-11-03

**Correctness:** 3
**Technical Novelty And Significance:** 2
**Empirical Novelty And Significance:** 2
**Recommendation:** 3
**Confidence:** 5

**Main Review:**

Strengths:
- the paper is well-written and easy to follow
- the paper considers a variety of attacks against federated learning, including both security and privacy attacks

Weaknesses:
- The paper lacks a discussion of the advantage of robustifying the algorithm at the client side over the server side in the introduction. The authors mention that the server-side defense changes the role of the central sever from being honest to trusted. Trusting the server might be a compromise on privacy. However, for security attacks, why should the clients trust the server to defend against the attacks given that the server is often an entity incentivized to train a high-performing model in practice?
- Section 3: it is mentioned that layer-wise bounds could reduce the variance but no experiments could be found to validate this point.
- Evaluation on Avaliability Poisoning attacks: The authors should show the federated learning with continuous gradient as a baseline.
- Evaluation on Integrity backdoor attacks: The experiments show some well-known defenses like Krum, FoolsGold, and Auror are not effective at all, which is a bit counter-intuitive. Any insights into why these defense do not work well? Is the finding generalizable to other datasets and model architectures?
- Discussion on Client Inference Attacks: The claim "However, a discrete local model cannot be used to infer any valuable information" is not sufficiently justified. Given the existing work on black-box model inversion attacks, like [1], one could even recover the training data from model output. It is reasonable to expect that some private information about training data can be inferred from discretized gradients in multiple rounds as it is more fine-grained information than model output.
[1]: https://arxiv.org/abs/1902.08552

**Summary Of The Paper:**

The paper proposes a federated learning algorithm that discretizes the client updates in order to achieve better robustness against various attacks. The contributions include (1) the development of the algorithm, (2) the theoretical analysis on the utility, robustness, and convergence of the algorithm and (3) empirical studies of the algorithm's robustness.

**Summary Of The Review:**

Discretization is widely used for reducing communication burden in federated learning. This paper is the first to consider its effect on robustness, which is an interesting idea. However, some of the claims made in the paper are not grounded. The evaluation of the algorithm is also limited. In some setting, it's not clear the advantage of the discretized gradient over continuous gradient; in other settings, the evaluation is limited to a specific dataset and hence it's not clear whether or not the findings are generalizable. Also, the privacy inference attack setting is not evaluated. So I think the current version of the paper does not meet the standard of ICLR.

---

### Official Review · Reviewer_QJJA · 2021-11-03

**Correctness:** 2
**Technical Novelty And Significance:** 2
**Empirical Novelty And Significance:** 2
**Recommendation:** 5
**Confidence:** 4

**Main Review:**

Strength
1.	Important topic.

2.	Using discretization seems to be an interesting idea.

3.	Well written.

Weakness:
1.	In Corollary 4.3.1, the theoretical robustness is based on “Let N be infinity” and “fixed F attackers”. This does not make sense to me. In this case, won’t existing FL methods be robust?

2.	The proposed FedDiscrete is obviously vulnerable to adaptive attacks. For instance, an attacker can upload manipulated l and u to the server in line 4, Algorithm 1. What if l=-infinity and u=infinity? They will then be chosen in line 5 and this will result in a non-sense model.

3.	It seems that the utility loss is large in some cases. For instance, in Figure 3, the accuracy may drop from 95% to 70%.

4.	Discretization has been applied to federated learning. For instance, signSGD [A] sends the sign of the gradients. I think the authors may need to discuss the existing methods and empirically compare with them. Also, stronger attacks and defenses are not compared.

[A]  signSGD: Compressed optimisation for non-convex problems. In ICML, 2018.

[B] A Little Is Enough: Circumventing Defenses For Distributed Learning. In NeurIPS, 2019.

[C] Local Model Poisoning Attacks to Byzantine-Robust Federated Learning. In USENIX Security Symposium, 2020.

[D] Provably Secure Federated Learning against Malicious Clients. In AAAI, 2021.

[E] CRFL: Certifiably Robust Federated Learning against Backdoor Attacks. In ICML, 2021.






**Summary Of The Paper:**

Federated learning (FL) has been shown to be vulnerable to weight poisoning attacks. An attacker who controls malicious clients can poison the clients’ model weights such that a backdoor to perform availability poisoning attacks, integrity backdoor attacks and inference attacks. In this work, the authors proposed a new FL algorithm called FedDiscrete, which probabilistically discretizes the clients’ model weights into two different values. The authors derived the convergence of FedDiscrete and empirically showed its performance against existing attacks. However, I am worried about the theoretical analysis on the robustness, as well as the empirical robustness against adaptive attacks.

**Summary Of The Review:**

Comparison with existing attacks and defenses is insufficient and adaptive attacks are not considered.  Theoretical analysis relies on strong assumptions.

---

### Decision · Program_Chairs · 2022-01-20

**Decision:**

Reject

**Comment:**

This manuscript proposes a quantization approach to improve adversarial robustness. Reviewers agree that the problem studied is timely and the approach is interesting. However, note concerns about the novelty compared to closely related work, the quality of the presentation, the strength of the evaluated attacks compared to the state of the art, among other concerns. There is no rebuttal.